# The Current Epidemiology of Urinary Incontinence and Urinary Tract Infections After Spinal Cord Injury—A Model Systems Spinal Cord Injury Examination (2016–2021)

**DOI:** 10.3390/jcm14051434

**Published:** 2025-02-21

**Authors:** Christopher Elliott, Evgeniy Kreydin, James Crew, Kazuko Shem

**Affiliations:** 1Santa Clara Valley Medical Center, San Jose, CA 95128, USA; james.crew@hhs.sccgov.org (J.C.); kazuko.shem@hhs.sccgov.org (K.S.); 2Department of Urology, Stanford University School of Medicine, Stanford, CA 94305, USA; 3Rancho Los Amigos National Rehabilitation Center, University of Southern California, Los Angeles, CA 90007, USA; evgeniy.kreydin@med.usc.edu

**Keywords:** urinary tract infection, urinary incontinence, spinal cord injury, neurogenic bladder, epidemiology

## Abstract

**Introduction**: Neurogenic bladder disorders after spinal cord injury (SCI) are often problematic with urinary incontinence (UI) and recurrent urinary tract infections (UTIs) contributing to patient morbidity. Our study objective was to provide a current snapshot of the frequency of UI and UTIs in the SCI population while also quantifying their association with hospitalization. **Methods**: The National Spinal Cord Injury Database (2016–2021) was accessed for persons within five years of injury. The self-reported frequency of UI in the last month (none, daily, weekly, monthly), the number of UTIs requiring antibiotic treatment, and the number of hospitalizations (including reason) in the prior year were evaluated. **Results**: Our cohort comprised 5106 individuals within 5 years of SCI. Approximately 40% reported UI in the past month and over 50% had a UTI requiring antibiotics in the past year. Incontinence was significantly more frequent in those performing clean intermittent catheterization (CIC) (52% overall, 17% daily) compared to indwelling catheters (29% overall) or volitional void (22% overall) (*p* < 0.001 for each). Conversely, UTIs were most common in those using indwelling catheters (79% with at least one UTI) or CIC (70%) compared to diapers/condom catheters (46%) or volitional void (19%) (*p* < 0.001 for each). Increasing UI and UTI occurrences were both associated with an increased frequency of hospitalization in the prior year. **Conclusions**: UI and UTIs are common problems after SCI. While differing frequencies of UI and UTI are present based on bladder management, the overall frequency suggests that a continued emphasis on prevention is needed to potentially increase quality of life and decrease hospitalization.

## 1. Introduction

For the estimated 200,000+ patients with a spinal cord injury (SCI) living in the United States, neurogenic bladder (NGB) dysfunction is often a significant source of morbidity and decreased quality of life [1,2,3]. This is often highlighted in studies where improved bladder function outcomes are given a higher priority than ever walking again. Amongst the most common sources of patient dissatisfaction are the four Is (urinary **I**ncontinence, urinary tract **I**nfection, bladder management **I**ndependence, and bladder management **I**nconvenience), with urinary incontinence and urinary tract infection (UTI) cited as the most bothersome [4,5]. Failing to alleviate urinary incontinence and UTIs, in addition to decreasing patient satisfaction, can also lead to additional morbidity, such as decubitus ulcer, sepsis, and hospitalization, all of which come at a substantial monetary cost [6].

In the last fifteen years, a number of therapeutic options have been added to assist with urinary incontinence treatment beyond anticholinergic medications and augmentation cystoplasty, such as beta-3 agonists and onabotulinum toxin A bladder injections [7,8,9]. Similarly, it is increasingly recognized that recurrent UTIs and febrile UTIs (which may cause urinary incontinence) may have contributing causes (such as genitourinary stones, poor bladder compliance, or poor bladder management) that are often diminished with a proper urologic intervention beyond antibiotics alone [10,11,12]. With the availability of new therapeutic options and an increased physiological understanding, one would expect that urinary incontinence and UTI rates would begin to improve; however, current epidemiologic data on this are sparse.

Our study objective was to provide a current snapshot of the frequency of UI and UTIs in the United States’ SCI population. In addition, we also investigate the association of UI and UTIs with hospitalization to further characterize the burden of health complications.

## 2. Methods

We assessed data from Form II of The National Spinal Cord Injury Database (NSCID) for the years 2016 to 2021, focusing on individuals with follow-up at 1 and 5 years after rehabilitation discharge [13]. The study period above was chosen as questions on the frequency of urinary incontinence were not introduced into the dataset until 2016. The NSCID dataset includes information on persons with traumatic SCI in designated Spinal Cord Injury Model Systems centers in the United States. Form II is a follow-up questionnaire that is administered one year after injury, five years after injury, and every five years thereafter. Within Form II, data on self-reported bladder management type, urinary incontinence in the last month (none, daily, weekly, monthly), UTI requiring antibiotic treatment in the past year (any, 1–2, 3–4, 5 or more), number of hospitalizations within the past year (up to eight), and the reason for hospitalization (with 17 distinct categories) are collected. Amongst the categories for hospitalization, UTI is included within “Diseases of genitourinary system, including renal, urethral, ureteral, and bladder stones, and conditions, urinary tract infections, diseases of the prostate, orchitis/epididymitis, disorders of genital organs, disorders of the breast and female pelvic organs”. All data on UI, UTIs, and hospitalization are self-reported and not confirmed with a hospital record review.

All individuals 13 years or older, who presented for 1- or 5-year follow-up, were included (7 persons excluded). In the rare case that an individual had both a 1- and 5-year follow-up over the period examined, we only included the most recent data collected for the purposes of this study. In addition to age <13 years old, our only other exclusion criterion was missing bladder management (n = 93) or an AIS (American Spinal Injury Association Impairment Scale) classification of E (no evidence of spinal cord injury at the time of rehabilitation discharge (n = 1)). Bladder management was based on self-report at the 1- or 5-year follow-up (volitional void, diaper/condom catheter, clean intermittent catheterization (CIC), indwelling catheter (urethral or suprapubic), or ileal conduit). In instances where more than one bladder management method is used, the NSCID only captures what the subject identifies as their primary bladder management method. Individuals using diapers or condom catheters were classified as having daily incontinence. Chi-square statistics were used to compare proportions between samples, *T*-tests were used to compare continuous variables, and multivariable logistic regression modeling including age (years), sex, race/ethnicity, bladder management, degree of urinary incontinence, UTI frequency, year of follow-up (1 vs. 5), AIS impairment, and level of injury (C,T, L/S) was used for determining associations with hospitalization in the past year. All data manipulation and statistics were performed using Stata version 12.1 (StataCorp, College Station, TX, USA).

## 3. Results

Over the study period (2016–2021), there were 5106 individuals with Form II NSCID data (2909 at 1 year and 2197 at 5 years post-rehabilitation discharge). The study cohort was predominantly male (79.0%) and White (~68.2%), with the most common site of injury being cervical (52.4%). Amongst bladder management methods at the time of follow-up, CIC was the most common (38.1%), followed by volitional void (33.9%), indwelling catheters (22.1%), and diapers/condom catheters (5.7%). Ileal conduits were rare (0.2%) in the study cohort. There was no significant difference in the makeup of the cohort based on the year of follow-up (Table 1).

### 3.1. Urinary Incontinence

In the overall cohort, urinary incontinence was present in 36.2%. This was equally divided amongst those leaking daily (11.0%), weekly (10.8%) or monthly (14.4%). There were no overt differences in the degree of leakage based on year of follow-up (1 vs. 5 years) (Table 1). When stratified by bladder management type, distinct differences in the frequency and degree of incontinence were seen. Those performing CIC were much more likely to report any urinary incontinence (51.8%) compared to indwelling catheters (29.0%) or volitional voiders (22.3%) (*p* < 0.001 for both comparisons). In the CIC subgroup, daily leakage (17.1%) accounted for roughly one-third of those reporting leakage, while in the indwelling catheter group, monthly leakage (16.9%) predominated, accounting for more than half of the reported urinary incontinence in the group (Table 2). Increased incontinence frequencies were also seen in females compared to males, those with increasing frequency of UTIs, and those with AIS A injuries. These univariate associations continued to be present in the multivariate analysis (Appendix A).

### 3.2. Urinary Tract Infection

Urinary tract infections were also common in the SCI cohort, with more than half (53.2%) of SCI patients reporting at least one UTI treated with antibiotics in the preceding 12-month period and 25.5% reporting three or more such UTIs during the same period (Table 1). When stratified by bladder management type, distinct differences were seen. Specifically, individuals relying on indwelling catheters (78.6%) or CIC (70.3%) were significantly more likely to have had a UTI requiring antibiotic treatment in the past year compared to those employing diapers/condom catheters (45.8%, *p* < 0.001 for both comparisons) and those able to volitionally void (19.2%, *p* < 0.001 for both comparisons). Those with indwelling catheters were also more likely to have had multiple UTIs, with a greater proportion of users experiencing three or more UTIs in the past 12 months (44.7%) compared to CIC users (33.2%) (*p* < 0.001) while also averaging more UTIs per year (indwelling catheters with an estimated 2.10 UTIs per year, comparted to CIC (1.63 UTIs/year), diapers/condom catheter (0.93 UTIs per year), or volitional void (0.34 UTIs per year) (*p* < 0.001 for each comparison)) (Table 2). Fewer UTIs were noted overall in year 5 after SCI compared to year 1 (52% vs. 58%, *p* < 0.001) and in also in those reporting no urinary incontinence (47.9% vs. 62.2%, *p* < 0.001). In addition, more frequent UTIs were found in females compared to males. These univariate associations continued to be present in the multivariate analysis (Appendix A).

### 3.3. Association with Hospitalization

Of the overall study population without missing hospitalization data (n = 4843), 32.1% had been hospitalized in the prior year, with 593 (38.1%) of the hospitalizations being due to a genitourinary condition. Both the increasing frequency of urinary incontinence and UTI were associated with an increase in the frequency of overall hospitalizations and hospitalization for genitourinary issues (Table 3). This association continued to be present in the multivariate analysis, where the presence of an indwelling catheter, one or more UTIs in the prior 12 months, and daily urinary incontinence were associated with an increased chance of hospitalization, while less complete injury and time since injury (5 years vs. 1 year) decreased the chance of hospitalization. The most striking association for chance of hospitalization was the number of UTIs in the prior year, with increasing numbers of UTIs associated with an increased chance of hospitalization in a gradated fashion (Table 4).

## 4. Discussion

Despite urological advances in the past quarter century, genitourinary problems in the SCI population continue to be highly prevalent. Over one-third of people with SCI in this study were found to have urinary incontinence occurring at least monthly. Similarly, UTIs were also common, with more than half of people with SCI reporting at least one UTI requiring antibiotics in the past year and ~25% reporting three or more UTIs in the past year. These problems appear to represent more than just simple patient inconvenience as both daily urinary incontinence and increasing numbers of UTIs were both associated with an increased frequency of hospitalization in the year prior (something that occurred in one-third of individuals with SCI).

Consistent with the existing literature, we found that bladder management method was closely associated with the frequency of urinary leakage and UTI [14]. Those performing CIC have the greatest burden of urinary incontinence (more than half leaking at least monthly and one in seven leaking on a daily basis), while those using either intermittent or indwelling catheters were the most likely to report a UTI (almost three-quarters having at least one UTI requiring antibiotics in the past year, and over one-third being treated for three or more urinary infections during the same time period). While some of these findings are intuitive, i.e., that and indwelling catheter would be associated with less urinary incontinence (as the bladder is always drained), others are not as obvious. Specifically, we found that bladder management was significantly associated with the frequency of hospitalization in the past year, with indwelling catheter use increasing the chance of hospitalization 2.5-fold compared to volitional voiders or those performing CIC.

While the initial costs of as SCI are estimated to range from USD 500,000 to 1,000,000 in the first year (with the majority being due to one’s initial hospitalization and rehabilitation) [15], the largest proportion of costs in subsequent years (which ranges on average from USD 75,000 to 210,000) is due to re-hospitalization [16]. Given that both urinary incontinence and UTI frequency are associated with an increase in hospitalization in those with SCI, our data potentially underscore the need for urologic attention when these conditions are present and the continued need for the development of novel treatments for both entities, whether in the form of new pharmaceuticals, UTI vaccines, or neuromodulation-type therapies (to name a few). With our data hinting at an association between urinary incontinence and UTI risk, perhaps ongoing urinary incontinence should be seen by all as a physiologic warning sign of bladder conditions (i.e., poor compliance, detrusor overactivity, or inappropriate bladder management) that place an individual at increased risk of future genitourinary infection and possible hospitalization. This notion has been supported in prior studies showing that intradetrusor onabotulinum toxin A injections for urinary incontinence can decrease UTI rates [17,18].

Whether or not aggressive intervention for urinary incontinence and UTI lead to a cost savings in terms of decreased hospitalization, the high prevalence and incidence of urinary incontinence should continue to be a priority in the SCI population due to its effects on overall quality of life (with bladder dysfunction being one of the primary drivers of patient dissatisfaction) [1,2,3]. Patients with SCI often cite urinary incontinence and UTIs as the primary reasons to change bladder management, likely in the hopes that their situation might improve [5,19]. In addition, decreases in overall socialization, return to work, and patient confidence further hinder one’s ability to live their best life due to their bladder disorder.

Our data are in line with a prior metanalysis, from a decade earlier, that found a similar overall frequency of urinary incontinence in the SCI population (52%) [20,21]. The fact that the frequency of incontinence has held steady despite FDA approval of both beta-3 agonist medications (vibegron and mirabegron) and intradetrusor onabotulinum toxin A injections for neurogenic bladder indications (to complement existing anticholinergic therapies) suggests that further work is required to either improve access to these agents or to facilitate the development of new therapies. We also show a similar frequency of UTIs in the current study compared to both a large prospective study of SCI patients followed by the Neurogenic Bladder Research Group (n = 1479) and a recent systematic review, each of which demonstrate an expected incidence of 0–3 urinary infections per year [22,23]. Similarly to these investigations, we also demonstrate that indwelling catheter users have a higher frequency of treatment for UTIs compared to intermittent catheterization, condom catheterization, or volitional voiding (in decreasing numbers) [14,22,23].

While our study is strengthened by the fact that we provide data on over 5000 individuals with SCI, there are certain limitations as well. One of the primary weaknesses is that the data are self-reported, and we do not have granular information on whether an individual received therapy for their urinary incontinence (anticholinergic medication, beta-3 agonist medication, onabotulinum toxin A injections, neuromodulation, surgery), what barriers might exist to receiving treatment (i.e., insurance approval, geographic proximity to a specialist), nor if they are seeing a specialist for their condition. Likewise, we do not have information on the type of urinary incontinence (stress versus urge), the degree of incontinence (number of pads or diapers used per day), nor the degree of bother that the leakage causes. Similarly, for those with UTIs, we do not have specific urine culture data or associated symptoms to ensure that a “true” UTI was present; rather, we only account for those patient-reported events that lead to the use of antibiotics, and for patients who are hospitalized, the degree of illness is difficult to ascertain as the specific reason for hospitalization (i.e., sepsis vs. low-grade fever vs. antimicrobial resistance requiring parenteral antibiotic) is not present. Future efforts to drive the universal use of a specific definition of a UTI in the neurogenic bladder population to reduce study heterogeneity and reduce the potential over-treatment of the SCI population with antibiotics would be welcome. Lastly, while most hospitalizations for genitourinary reasons in the SCI population are due to urinary tract infections (>90%), we do not know which individuals were hospitalized for a genitourinary infection versus kidney stones, bladder stones, or other genitourinary conditions.

Despite these limitations, we provide a current epidemiologic snapshot of the frequency of urinary incontinence and treatment of UTIs with antibiotics in the SCI population. Compared to other studies, our study population is significantly larger than other single-institution series, or even the aggregate summarized in a metanalysis that covered urinary incontinence studies up to the year 2010 (n = 519) [21]. Our study can also be relied on to be specific (as all patients in the Model Systems SCI cohort are diagnosed and treated in a spinal cord rehabilitation center of excellence) compared to claims-based analyses that rely on diagnostic codes which are unreliable and often overly inclusive [24]. As a result, our epidemiologic investigation should provide a starting point for any researcher interested in the topic of urinary incontinence and UTIs in the SCI population. In addition, our data, which suggest that the frequency of urinary incontinence and UTIs are potentially unchanged over time (despite new therapies), are hopefully a catalyst to evaluating our current approaches to these entities, with the thought that a new approach might be needed.

## 5. Conclusions

Urinary incontinence and UTIs are common problems after SCI. In addition to being potentially bothersome, both are associated with an increased chance of potentially costly hospitalizations and morbidity. While differing frequencies of UI and UTIs are present based on bladder management, the overall frequency suggests that a continued emphasis on prevention is needed to potentially increase quality of life and decrease hospitalization.

## Figures and Tables

**Table 1 jcm-14-01434-t001:** Model system spinal cord injury cohort characteristics (2016–2021).

		1-Year Follow-Up n = 2909	5-Year Follow-Up n = 2197
Median age		41	43
Sex			
	Male	2270 (78%)	1762 (80.2%)
	Female	637 (21.9%)	434 (19.8%)
	Other	2 (0.1%)	1 (0.1%)
Race			
	White	1925 (66.2%)	1557 (70.9%)
	Black	707 (24.3%)	473 (21.5%)
	Asian/Pacific Islander	81 (2.8%)	45 (2.1%)
	Other	157 (5.4%)	97 (4.4%)
	Unknown	39 (1.3%)	25 (1.1%)
Level of injury			
	Cervical	1521 (52.3%)	1154 (52.5%)
	Thoracic	1055 (36.3%)	746 (34.0%)
	Lumbar/Sacral	333 (11.5%)	297 (13.5%)
AIS class			
	A	879 (30.2%)	752 (34.2%)
	B	335 (11.5%)	288 (13.1%)
	C	419 (14.4%)	333 (15.2%)
	D	970 (33.3%)	666 (30.3%)
	Unknown	306 (10.5%)	158 (7.2%)
Bladder management			
	Volitional Void	986 (33.9%)	744 (33.9%)
	Diaper/Condom Catheter	183 (6.3%)	110 (5.0%)
	CIC	1095 (37.6%)	850 (38.7%)
	Indwelling Catheter	642 (22.1%)	486 (22.1%)
	Ileal Conduit	3 (0.1%)	7 (0.3%)
Number of UTI requiring antibiotics in the past year			
	None	1222 (42.0%)	1054 (48.0%)
	1–2	789 (27.1%)	546 (24.9%)
	3–4	444 (15.3%)	282 (12.8%)
	5 or more	301 (10.4%)	209 (9.5%)
	UTI, but unknown frequency	11 (0.4%)	7 (0.3%)
	Unknown	142 (4.9%)	99 (4.5%)
Urinary incontinence frequency in the past month			
	None	1670 (57.4%)	1321 (60.1%)
	Daily	295 (10.1%)	219 (10.0%)
	Weekly	305 (10.5%)	200 (9.1%)
	Monthly	393 (13.5%)	284 (12.9%)
	Unknown	246 (8.4%)	173 (7.9%)
Hospitalized in the past year *		961/2749 = 35.0%	594/2094 = 28.3%
Hospitalized for a genitourinary reason in the past year **		379/2739 = 13.8%	214/2084 = 10.2%

* Unknown = 160 (5.5%) at 1-year follow-up and 103 (4.7%) at 5-year follow-up. ** Unknown = 170 (5.8%) at 1-year follow-up and 113 (5.1%) at 5-year follow-up.

**Table 2 jcm-14-01434-t002:** Frequency of urinary incontinence and urinary tract infection based on bladder management in the model systems spinal cord injury cohort (2016–2021) (year 1 and year 5 follow-up cohorts combined).

	Volitional Void	Indwelling Catheter	CIC	Diaper/Condom Cath
**Frequency of urinary** **incontinence ***	n = 1650	n = 971	n = 1792	n = 293
Any	368 (22.3%)	282 (29.0%)	928 (51.8%)	-
Daily	106 (6.4%)	50 (5.2%)	306 (17.1%)	-
Weekly	115 (7.0%)	68 (7.0%)	291 (16.2%)	-
Monthly	147 (8.9%)	164 (16.9%)	331 (18.5%)	-
**UTI treated with antibiotics in the previous year ****	n = 1665	n = 1078	n = 1836	n = 277
Any	320 (19.2%)	847 (78.6%)	1290 (70.3%)	127 (45.8%)
1–2	226 (13.6%)	357 (33.1%)	670 (36.5%)	79 (28.5%)
3–4	64 (3.8%)	258 (23.9%)	374 (20.4%)	29 (10.5%)
5 or more	30 (1.8%)	225 (20.8%)	236 (12.9%)	18 (6.5%)
Have UTI but unknown frequency	0	7 (0.7%)	10 (0.5%)	1 (0.4%)
**Avg UTI per year**	0.34	2.10	1.63	0.93
**Any hospitalization in the past year**	22.1%	57.5%	34.8%	37.2%
**Hospitalization for GU** **reason in the past year**	2.9%	27.7%	11.8%	12.9%

* Missing urinary incontinence data: volitional void n = 80 (4.6%), indwelling catheter n = 157 (13.9%), CIC n = 153 (7.9%). ** Missing UTI data: volitional void n = 65 (3.8%), indwelling catheter n = 50 (4.4%), CIC n = 109 (5.6%), diaper/condom catheter n = 16 (5.5%). Not included = ileal conduit (n = 10).

**Table 3 jcm-14-01434-t003:** Frequency of urinary incontinence and UTIs in relation to hospitalization and hospitalization for a genitourinary reason in the model system spinal cord injury cohort (2016–2021) (year 1 and year 5 follow-up cohorts combined).

	Any Hospitalization	Hospitalization for GU Reason
**Frequency of Urinary Incontinence**		
None	914/2991 (30.6%)	311/2938 (10.6%)
Monthly	235/677 (34.7%)	86/662 (13.0%)
Weekly	162/505 (32.1%)	54/497 (10.9%)
Daily	215/514 (41.8%)	96/503 (19.1%)
Missing	419	506
**Number of UTIs in the Prior Year**		
None	457/2276 (20.1%)	30/2226 (1.3%)
1–2	510/1335 (38.2%)	187/1300 (14.4%)
3–4	352/726 (48.5%)	180/706 (25.5%)
5 or more	301/510 (59.1%)	184/500 (36.8%)
Missing	259	374

**Table 4 jcm-14-01434-t004:** Factors associated with hospitalization in the model systems spinal cord injury cohort (2016–2021) (year 1 and year 5 follow-up cohorts combined).

		Univariate		Multivariate	
		OR (95%CI)	*p*-Value	OR (95% CI)	*p*-Value
Age at Injury (years)		1.01 (1.00–1.01)	0.001	1.01 (1.01–1.02)	<0.001
Sex	Male	Ref	-	Ref	-
	Female	1.05 (0.91–1.20)	0.534	0.89 (0.76–1.06)	0.194
Race/Ethnicity	White	Ref	-	Ref	-
	Black	0.96 (0.83–1.10)	0.519	1.09 (0.93–1.28)	0.305
	American Indian, Alaska Native	1.18 (0.65–2.16)	0.583	0.93 (0.45–1.91)	0.847
	Asian, Pacific Islander	0.66 (0.44–0.98)	0.038	0.50 (0.30–0.81)	0.005
	Other, Multiracial	0.93 (0.70–1.25)	0.641	0.81 (0.57–1.16)	0.254
Bladder Management at Follow-up	Volitional Void	Ref	-	Ref	-
	Condom Cath/Diapers	2.08 (1.58–2.74)	<0.001	1.43 (1.04–1.97)	0.027
	Indwelling Catheter	4.76 (4.04–5.61)	<0.001	2.54 (1.99–3.23)	<0.001
	CIC	1.88(1.62–2.17)	<0.001	1.02 (0.81–1.28)	0.900
	Conduit	2.34 (0.66–8.35)	0.189	1.82 (0.27–12.23)	0.541
	Other	1.22 (0.57–2.62)	0.615	0.86 (0.38–1.99)	0.719
Number of UTIs in the last year treated with antibiotics	No UTI	Ref	-	Ref	-
	1 or 2 UTIs	2.46 (2.11–2.86)	<0.001	2.01 (1.69–2.39)	<0.001
	3 or 4 UTIs	3.75 (3.14–4.48)	<0.001	2.81 (2.28–3.46)	<0.001
	5 or more UTIs	5.73 (4.67–7.03)	<0.001	4.45 (3.52–5.63)	<0.001
Urinary Incontinence	None			Ref	-
	Daily incontinence	1.63 (1.34–1.97)	<0.001	1.52 (1.23–1.88)	<0.001
	Weekly incontinence	1.07 (0.88–1.31	0.493	0.98 (0.79–1.23)	0.869
	Monthly incontinence	1.21 (1.01–1.44)	.036	1.01 (0.84–1.23)	0.888
Year of Follow-up	1 Year	Ref	-	Ref	-
	5 Year	0.74 (0.66–0.83)	<0.001	0.72 (0.63–0.83)	<0.001
AIS Class	A	Ref	-	Ref	-
	B	0.89 (0.74–1.07)	0.223	0.85 (0.68–1.06)	0.147
	C	0.71 (0.59–0.84)	<0.001	0.74 (0.60–0.92)	0.008
	D	0.45 (0.38–0.52)	<0.001	0.70 (0.55–0.89)	0.003
Injury Level	Cervical	Ref	-	Ref	-
	Thoracic	0.97 (0.86–1.10)	0.673	1.00 (0.84–1.20)	0.967
	Lumbosacral	0.80 (0.60–1.20)	0.876	0.95 (0.70–1.31)	0.755

## Data Availability

The original contributions presented in the study are included in the article (and Appendix A), further inquiries can be directed to the corresponding authors.

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
