# Peer review of "The Current Epidemiology of Urinary Incontinence and Urinary Tract Infections After Spinal Cord Injury—A Model Systems Spinal Cord Injury Examination (2016–2021)"

_jcm, 2025, doi:10.3390/jcm14051434_

Round 1

Reviewer 1 Report

Comments and Suggestions for Authors

Excellent work highlighting the new data from the SCI registry. The results are in line and support all the prior work in this area but it is very robust data set and excellent analysis to highlight a major issue in SCI bladder management. 

This is a well written paper and I do not have any specific comments/edits or concerns. 

Author Response

We thank the reviewer for their kind comments.

Reviewer 2 Report

Comments and Suggestions for Authors

a well written manuscript and the authors correctly have outlined a number limitations and encourage others to take up research on these conditions

Author Response

(The authors gave the same response as above.)

Reviewer 3 Report

Comments and Suggestions for Authors

Dear Authors,

I have reviewed your manuscript on the epidemiology of urinary incontinence and UTIs following spinal cord injury, which effectively utilizes the National Spinal Cord Injury Database (2016-2021). While the paper presents valuable insights, I would like to suggest the following revisions to enhance its academic significance and practical utility:

  1. Abstract Section
  • Include study limitations
  • Provide specific numerical data for key findings
  • Strengthen practical recommendations in conclusions
  1. Introduction Section
  • Incorporate recent global epidemiological data on post-SCI UTIs and urinary incontinence
  • Clarify research objectives
  • Elaborate on the study's novelty and significance
  1. Methods Section
  • Detail data collection protocols
  • Specify inclusion/exclusion criteria
  • Elaborate statistical analysis methodologies
  • Address ethical considerations
  1. Results Section
  • Present UTI classification data
  • Include severity distribution analysis
  • Add subgroup analyses
  • Enhance visual representation of data (graphs, tables)
  1. Discussion Section
  • Expand comparative analysis with existing literature
  • Address study limitations comprehensively
  • Provide specific clinical practice recommendations
  • Suggest future research directions
  1. Conclusion Section
  • Strengthen practical implications
  • Summarize key findings concisely
  • Outline future research priorities
  1. References Section
  • Update with recent relevant literature
  • Include international research citations

These modifications would significantly enhance the manuscript's comprehensiveness and value to the scientific community. Your paper addresses an important topic, and these suggested revisions aim to maximize its impact.

Sincerely,

Author Response

We thank the reviewer for their kind comments; however, we struggle to understand what specifically the reviewer would like done as there are no “concrete” suggestions.  Rather, we have been given a laundry list of ambiguous tasks, most of which have been addressed in some way shape or form in our current manuscript.  Without more concrete suggestions (which we would typically give to others when performing manuscript reviews), we are unable to specifically respond further.

Reviewer 4 Report

Comments and Suggestions for Authors

This is a very interesting original article in a not so frequently reported topic. The authors must be complimented for the very detailed presentation of the methodology of their study, the large sample size (5106 individuals) and the clear language which fascilitates readers to understand the topic. The major drawback is the retrospective design, which is based on a database. Comments for the authors:

1) Please provide the affiliations after the title and the authors names.

2) What does the current article present is the prevalence of urinary incontinence and urinary tract infections in a very specific cohort of patients (spinal cord injury). However, the authors do not provide any informations about the management (pharmacological and surgical one) trends in this cohort. I would highly encourage the authors to provide such data, if they are available.

3) The authors should be more specific on the kind of urinary incontinence that their cohort had.

4) I would suggest to provide some information (literarure review) regarding the management of these patients in the discussion section.

5) Please provide authro contributions and the statements after the manuscript. Informed consent should have been given by the patients.

Author Response

1) Please provide the affiliations after the title and the authors names. – this information has been provided to the journal.

2) What does the current article present is the prevalence of urinary incontinence and urinary tract infections in a very specific cohort of patients (spinal cord injury). However, the authors do not provide any informations about the management (pharmacological and surgical one) trends in this cohort. I would highly encourage the authors to provide such data, if they are available. – Unfortunately, this data is not captured in the Model Systems database.  This is currently included within the limitations section of our manuscript as we agree that this information would have been interesting to study.

3) The authors should be more specific on the kind of urinary incontinence that their cohort had.

Again, this data (stress vs urge vs other type of incontinence) is not captured in the Model Systems database.  This limitation is currently included in the limitations section of our manuscript as we agree that this information would have been interesting to study.

4) I would suggest to provide some information (literarure review) regarding the management of these patients in the discussion section.

A brief discussion of common methods to treat urinary incontinence (altered bladder management, pharmacotherapy, intradetrusor abotulinum toxin injection) is currently in the 2nd paragraph of the manuscript Introduction and perhaps overlooked by the reviewer?  Likewise similar information for UTI treatment (antibiotics, altered bladder management, removal of bladder stones) is also present in this section of the manuscript.  Should this not suffice, please advise us further.

5) Please provide authro contributions and the statements after the manuscript. Informed consent should have been given by the patients.

This information has been provided to the journal.

Round 2

Reviewer 3 Report

Comments and Suggestions for Authors

I have no additional comments to make.

Author Response

nothing to respond to

Reviewer 4 Report

Comments and Suggestions for Authors

The authors have not provided a literarure review of the management of patients with spinal cord injury and incontinence or UTIs, as requested. What the authors provide in the second paragraph of their introduction is a very short discussion of incoentince in general. What I would highly recommend authors is to search the literature for the threrapeutic challenges of patients with spinal cord injuries.

Author Response

nothing to respond to